# GRADIENT-BASED OPTIMIZATION OF NEURAL NETWORK ARCHITECTURE

**Will Grathwohl, Elliot Creager, Seyed Kamyar Seyed Ghasemipour, Richard Zemel** [*]
Department of Computer Science
University of Toronto
Vector Institute
{wgrathwohl,creager,kamyar,zemel}@cs.toronto.edu

## ABSTRACT

Neural networks can learn relevant features from data, but their predictive accuracy and propensity to overfit are sensitive to the values of the discrete hyperparameters that specify the network architecture (number of hidden layers, number of units per layer, etc.). Previous work optimized these hyperparmeters via grid search, random search, and black box optimization techniques such as Bayesian optimization. Bolstered by recent advances in gradient-based optimization of discrete stochastic objectives, we instead propose to directly model a distribution over possible architectures and use variational optimization to jointly optimize the network architecture and weights in one training pass. We discuss an implementation of this approach that estimates gradients via the Concrete relaxation, and show that it finds compact and accurate architectures for convolutional neural networks applied to the CIFAR10 and CIFAR100 datasets.

## 1 INTRODUCTION

Neural networks are a composition of many simple components that form a very powerful function approximator. The structure of this composition is typically called an "architecture." While simple to describe at a high level, the space of neural network architectures is prohibitively large to search over. The performance of a neural network on a given task is very sensitive to the choice of architecture. Moreover, because the architecture hyperparmeters (e.g., width, depth) are discrete, they cannot be optimized along with the weights of the model using backpropagation.

Our aim is to automatically incorporate architecture search into the neural network optimization pipeline. Previous work in this space treats architecture as a hyperparameter akin to learning rate and regularization strength, which can be optimized by grid search or random search (Bergstra & Bengio, 2012). Snoek et al. (2012) fits a Gaussian process to (hyperparameter, validation loss) pairs and seeks to optimize this unknown function constrained to an exploration budget.

We instead treat the architecture as a parameter similar to the weights of the network and use gradient-based optimization to learn both simultaneously, thus removing architecture from the set of hyperparameters which must be searched over.

## 2 VARIATIONAL MODEL OPTIMIZATION

We phrase this problem within the framework of variational optimization (Staines & Barber, 2012). Consider a supervised learning setup where we are given dataset $\mathcal{D}$ of input-label pairs $(x, y)$. We wish to predict the labels $y$ with a flexible classifier (e.g., neural network). We seek to find weights $w$ which minimize

$$\mathcal{L}(w) = \mathop{\mathbb{E}}_{x,y\sim\mathcal{D}} \left[ \ell(f(x, w), y) + \lambda ||w||_2^2 \right],\tag{1}$$

where $\ell(\cdot, \cdot)$ is a suitable loss function and $\lambda$ is a hyperparameter specifying the amount of weight decay added to the objective. Assuming a fixed architecture, standard neural network training proceeds via gradient-decent on $w$.

---

[*] The first 3 authors contributed equally to this work.

Letting $A$ denote the discrete specifications of the network architecture (depth, width), we seek to find $A$ and $w$ which minimize

$$\mathcal{L}(w, A) = \mathbb{E}_{x,y \sim D} \left[ \ell(g(x, w, A), y) + \lambda ||w||_2^2 + \gamma r(A) \right], \tag{2}$$

where $r(A)$ is a regularization function, weighted by $\gamma$, that penalizes the complexity of a model architecture. Note that the prediction $g(x, w, A)$ takes into account the choice of architecture, and is therefore only defined for a discrete set of points.

**Variational Optimization**   The global minimum of an objective $\mathcal{L}$ over a set of parameters $\theta$ is upper bounded by the expectation of that objective under a distribution $q$ on $\theta$ with suitable support:

$$\min_{\theta} \mathcal{L}(\theta) \leq \mathbb{E}_{\theta \sim q(\theta)} [\mathcal{L}(\theta)] \tag{3}$$

Given this fact, the variational optimization (VO) framework seeks to minimize $\mathcal{L}(\theta)$ by optimizing the distribution $q$ in order to minimize the expectation on the RHS of the above inequality.

VO is directly applicable to our optimization problem: Whereas directly solving Equation 2 involves a discrete optimization over $A$, we can instead place a parametric distribution $q_\phi$ over architecture choices $A$ and optimize the right hand side of

$$\min_{w, A} \mathcal{L}(w, A) \leq \mathbb{E}_{A \sim q_\phi(A)} [\mathcal{L}(w, A)] \triangleq U(w, \phi). \tag{4}$$

As $(w, \phi)$ approach their optimal values $(w^*, \phi^*)$, the approximation gap closes and $q_\phi$ places nearly all of its probability mass on the globally optimal $A^*$.

We refer to this optimization problem as variational model optimization (VMO) and use gradient-based optimization to minimize the objective $U(w, \phi)$. Deriving a gradient-based VMO algorithm involves specifying a regularization function $r(A)$ over architectures, a sufficiently expressive distribution $q_\phi(A)$, and an estimator for the gradients of $U(w, \phi)$.

## 3   VARIATIONAL MODEL OPTIMIZATION FOR NEURAL NETWORKS

We restrict our attention to optimizing for the depth and width of a discriminative neural network with cross entropy loss. In standard approaches to classification, the final layer's activations $z_D$ parameterize a categorical distribution over predictions as $p(c_i|x) \propto \exp z_{D_i}$.

**Depth Optimization**   When optimizing the depth of a model, we set a maximum allowable depth $D$ and seek to find the optimal depth $d^*$ during training ($1 \leq d^* \leq D$). To do so, we construct a neural network with $D$ layers and build a classification layer on top of each intermediate layer's activations $a_d$. A classification layer at layer $d$ is a linear projection mapping $a_d$ to logits $z_d$. The variational distribution over model depth is taken to be a categorical distribution $q_\phi(d)$ with parameters $\phi$, and we represent samples $d \sim q_\phi$ by one-hot vectors $\texttt{one\_hot}(d)$. When depth $d$ is sampled from $q_\phi$, the model's output logits are

$$\hat{z} = \sum_{i=1}^{D} \texttt{one\_hot}(d)_i \cdot z_i \tag{5}$$

We define the regularization function over depth to be $r(d) = d$.

**Width Optimization**   To optimize the width of a neural network's layers, we consider models with $\{1, \ldots, K_l\}$ features in each layer $l$. The model is implemented as a neural network with $K_l$ features in the $l$th layer. Models with $k_l < K_l$ features are generated by multiplying the layer's activations $a_l$ with a binary mask, $m_l = [\underbrace{1, \ldots, 1}_{k_l}, \underbrace{0, \ldots, 0}_{K_l - k_l}]$ and passing the result $m_l \cdot a_l$ as input to the next layer.

We place a categorical distribution $q_{\psi_d}$ over the width of each layer, and represent the joint distribution over all layers using a fully factorized distribution $q_\psi(k_1, \ldots, k_D) = \prod_{l=1}^{D} q_{\psi_l}(k_l)$. Samples from $q_{\psi_l}(k_l)$ are encoded as 1-hot vectors $\texttt{one\_hot}(k_l)$ and $m_l$ can be produced as

$$(m_l)_i = \sum_{j \geq i} \texttt{one\_hot}(k_l)_j. \tag{6}$$

| Dataset | Model | Accuracy | Depth | Width |
|---------|-------|----------|-------|-------|
| Cifar10 | WRN-28-10 | 96.00% | 28 layers | 100% |
| Cifar10 | Depth | 96.01% | 26 layers | 100% |
| Cifar10 | Width | 95.22% | 28 layers | $81.89\% \pm 12.51\%$ |
| Cifar10 | Joint | 95.00% | 28 layers | $77.14\% \pm 12.72\%$ |
| Cifar100 | WRN-28-10 | 80.75% | 28 layers | 100% |
| Cifar100 | Depth | 80.21% | 26 layers | 100% |
| Cifar100 | Width | 77.15% | 28 layers | $75.98\% \pm 17.58\%$ |
| Cifar100 | Joint | 76.23% | 28 layers | $77.83\% \pm 17.03\%$ |

Table 1: Results on Cifar10 and Cifar100. Median accuracy over 5 runs.

We define our regularization function to be $r(k_1, \ldots, k_D) = \sum_{l=1}^{D} k_l$.

**Joint Optimization**    To optimize width and depth we combine both of the above models. We place a distribution over depth and width as $q_{(\phi,\psi)}(d, k_1, \ldots, k_D) = q_\phi(d) \cdot \prod_{l=1}^{D} q_{\psi_l}(k_l)$ and proceed as defined above. We define the regularization function as $r(d, k_1, \ldots, k_D) = d + \sum_{l=1}^{D} k_l$.

**Concrete Gradient Estimation**    The Concrete relaxation replaces the discrete categorical distribution with a continuous distribution that admits the reparameterization gradient estimator (Kingma & Welling, 2013). Samples from the Concrete relaxation of a multinomial distribution with parameter $\phi$ can be generated as softmax$((\log \phi - \log(-\log u))/t)$, where $u \sim$ uniform$[0,1]^{|\phi|}$. $t$ is a "temperature" parameter which controls the bias-variance trade-off of the estimator. Samples from this distribution can replace any 1-hot samples and then the reparameterization estimator can be used to estimate the gradients of the samples with respect to $\phi$. We replace the 1-hot samples in Equations 5 and 6 with samples from the Concrete relaxation.

## 4    EXPERIMENTS

We experiment with VMO on the challenging task of training Wide Resnets (Zagoruyko & Komodakis, 2016) on Cifar10 and Cifar100 (Krizhevsky et al.). Our models have the same base architecture as WRN-28-10. The depth model optimizes the number of blocks and the width model optimizes the width of each block[1]. We anneal the temperature $t$ of the Concrete relaxation from 0.1 to 0 throughout training on a log-scale. We use a regularization penalty of $10^{-3}$ for depth and $10^{-8}$ for width. These hyperparameter values were chosen via cross-validation. All other experimental details are as in Zagoruyko & Komodakis (2016). Cf. Table 1 for a summary of our learned architectures and their performances. On both Cifar10 and Cifar100, the depth model chose an architecture 1 block smaller than that of the baseline WRN model and on Cifar10 this resulted in a slight increase in accuracy with an $\sim 8\%$ reduction in parameters. While we noticed a decrease in performance with the width a and joint models, we observed a considerable reduction in the number of parameters while maintaining near state-of-the-art accuracy.

## 5    CONCLUSION

We have presented a method to jointly optimize a neural network's architecture along with its weights. This method utilizes variational optimization to accomplish this task and introduces minimal computation overhead compared to standard neural network training. While we feel the approach is promising there are a number of issues that must be dealt with for it to achieve greater success. We believe that most of these issues are due to the bias introduced by the Concrete relaxation. Empirically, the biased estimators increase the size of the generalization gap. We also believe that the exploration caused by the variational distribution slows convergence of the weights. We believe better results could be obtained by training the VMO models longer after the VO procedure has approximately made its architecture choice.

---

[1] Blocks are as is defined as in  Zagoruyko & Komodakis (2016).

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

## 6 APPENDIX

### 6.1 ALTERNATIVE GRADIENT ESTIMATORS

The variational objective $U(w, \phi)$ (Equation 4) consists of an expectation over discrete model choices $A$. Hence, the variational distribution parameters $(\phi, \psi)$ cannot be optimized directly with backpropagation and we must estimate the gradients with respect to these parameters. The score function estimator (Williams, 1992) could be used, but it is known to have prohibitively high variance. To remedy this, we could utilize the lower-variance, unbiased estimators of Tucker et al. (2017) and Grathwohl et al. (2017) which utilize control variates to reduce the variance of the score function estimator. We initially experimented with these methods but found them difficult to scale to state of the art convolutional neural networks due to memory and computational constraints.

For that reason we have decided to utilize the low-variance, but biased Concrete relaxation (Maddison et al., 2016; Jang et al., 2016) to estimate the gradients of our objective. We believe that the bias these estimators add is responsible for most of the decrease in performance we observe when compared to the baseline WRN models.

We believe improvements could be achieved with unbiased gradient estimators or a different temperature annealing strategy.

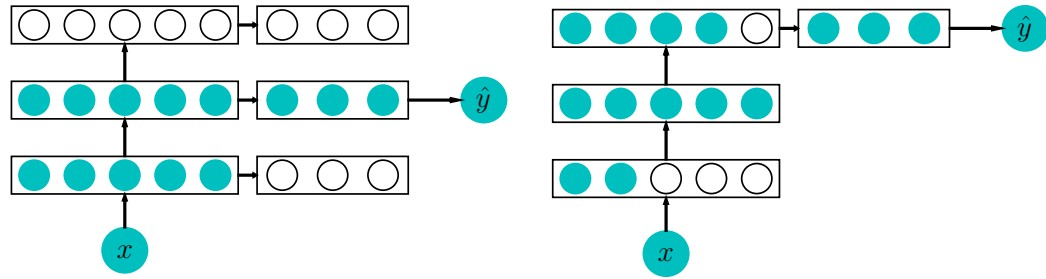

(a) During depth optimization sampling a model corresponds to deciding the depth at which to classify.

(b) During width optimization sampling a model corresponds to sampling a number of active neurons per layer.

Figure 1: In variational model optimization we minimize the expected loss over a distribution of models. 1a and 1b show single samples from the distributions over models in depth and width optimization, respectively.

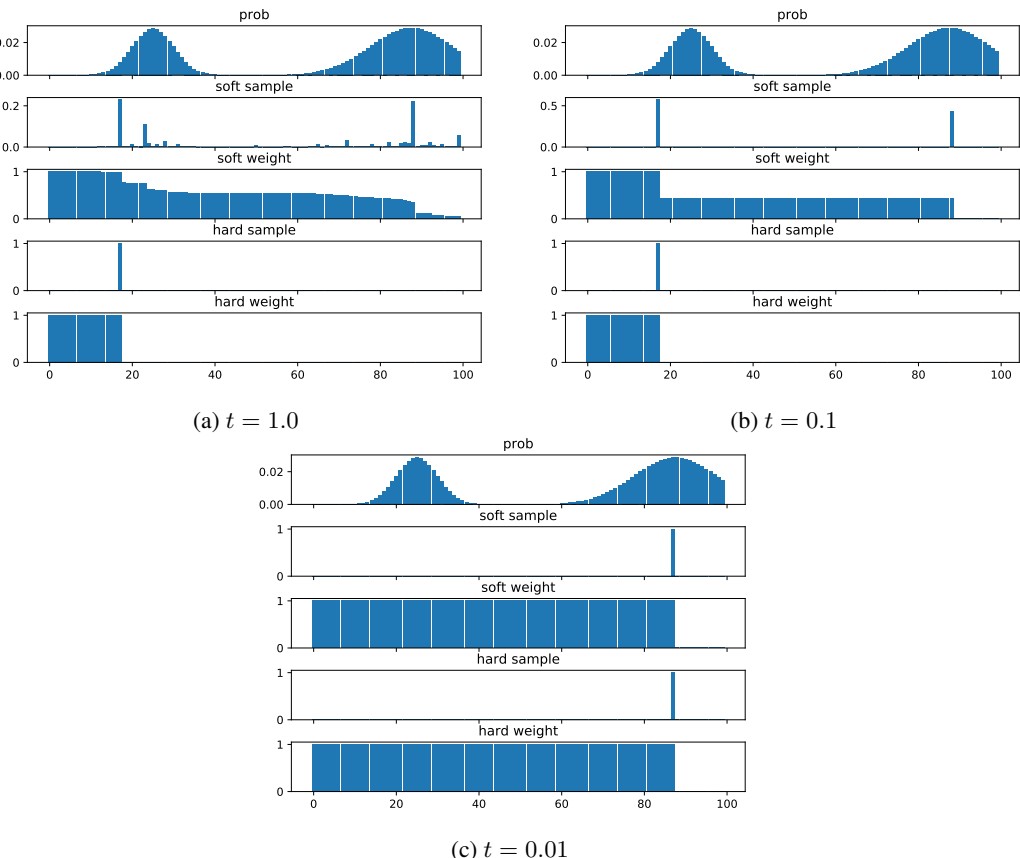

Figure 2: Concrete sampling procedure for width prediction within one layer of the network for temperature $t \in \{1.0, 0.1, 0.01\}$. The first row shows the Categorical distribution over the width to choose for the layer. The fourth and fifth row corresponds to samples and masks $m$ from the Categorical distribution as described in Equation 6, which we use at test time. The second and third row corresponds to samples and masks $m$ from the Concrete relaxation with temperature $t$, which we use during training. The soft masks approaches the hard masks in the limit of $t \to 0$.

## 7 Continuous Relaxation for Choice of Width

In training width models, we make use of concrete gradient estimation and a continuous relaxation of Equation 6. Denoting the activations at layer $l$ by $[a_1, ..., a_{K_l}]$, the continuous approximation for choosing a certain width is given as follows: Given a multinomial distribution over the width to choose, we take a sample $\nu$ using the concrete relaxation with temperature parameter $t$. The final output of this layer is formed to be $[m_1 \cdot a_1, ..., m_{K_l} \cdot a_{K_l}]$ where $m_i = \sum_{j \leq i} \nu_j$. As the $t$ approaches 0, $m_i$ approach the hard values from Equation 6. Samples of $m_i$ for different temperatures are presented in 2.

