# OpenReview forum: "Gradient-based Optimization of Neural Network Architecture"
_ICLR.cc/2018/Workshop — Accept_

### Official Review · AnonReviewer2 · 2018-02-23
**Interesting research direction**

**Rating:** 9
**Confidence:** 5

**Review:**

The proposed approach for inferring key configuration aspects of deep networks is novel and sound. Even though the provided results correspond to very early stage developments, they are worth of presentation in ICLR workshops.

---

### Official Review · AnonReviewer1 · 2018-03-12
**Interesting small scale work on learning neural network architectures**

**Rating:** 6
**Confidence:** 4

**Review:**

This paper looks at formulating a learning to learn method for selecting the depth and width of neural networks. This problem is first formulated as a variational optimization problem: to pick an architecture A to minimize the loss function L(w, A), we instead look at minimizing this in expectation, over a distribution q over A. To pick depth, a linear classifier is trained for each layer (there is a missing cite to the 'linear classifier probes' paper) and element wise multiplied with the sample from the distribution q_phi. Width is done by having masks of form [1,...,1,0...,0] instead of the linear classifiers.

The setting of the problem is introduced clearly, but some of the details could use additional clarification. For example, in equation (5), if d is a one hot vector, then only one of the z_i's contributes to \hat{z}, which is a little misleading with the summation term.

This work is a nice example of a simple way to study learning to learn, and would be well suited to the ICLR workshop. However, there are important directions to address before becoming a full fledged paper -- e.g. how expressive are the q_\phi, is the regularization function really necessary?

Most importantly, can this method find architectures that are nothing like the starting point? Currently, this just appears to apply small changes to the starting architecture. It's not bad for a workshop submission, but much more would have to be done for a conference paper.

---

### Decision · Program_Chairs · 2018-03-20
**ICLR 2018 Workshop Acceptance Decision**

**Decision:**

Accept

**Comment:**

Congratulations, your paper was accepted to the ICLR workshop.